# On Reducing Action Labels in Planning Domains

**Harsha Kokel, Junkyu Lee, Michael Katz, Kavitha Srinivas, Shirin Sohrabi**

IBM Research

{harsha.kokel,Junkyu.Lee,michael.katz1,kavitha.srinivas}@ibm.com, ssohrab@us.ibm.com

## Abstract

Planning tasks succinctly represent labeled transition systems, with each ground action corresponding to a label. This granularity, however, is not necessary for solving planning tasks and can be harmful, especially for model-free methods. In order to apply such methods, the label sets are often manually reduced. In this work, we propose automating this manual process. We characterize a *valid* label reduction for classical planning tasks and propose an automated way of obtaining such valid reductions by leveraging lifted mutex groups. Our experiments show a significant reduction in the action label space size across a wide collection of planning domains. We demonstrate the benefit of our automated label reduction in two separate use cases: improved sample complexity of model-free reinforcement learning algorithms and speeding up successor generation in lifted planning. The code and supplementary material are available at https://github.com/IBM/Parameter-Seed-Set.

## 1 Introduction

AI Planning tasks, described in the planning domain description language (PDDL) (McDermott 2000), induce transition graphs with states as nodes and transitions between states as labeled edges. These labeled transition systems (LTS) feature a unique label for each ground action. They identify transitions induced by the same action on different states with the *same label*. In practice, these labels are primarily used to distinguish *applicable* operations in a given state. So, a much smaller, sufficient set of labels might be attainable. Consider a gripper domain (McDermott 2000) where a robot moves balls between two rooms. Figure 1 depicts a PDDL task in this domain. Consider the lifted action `pick` and its second parameter `?r:room`. All applicable groundings of the action `pick` in any given state will have the same value for `?r`, namely the current room the robot is in. Therefore, this parameter is not essential for distinguishing LTS transitions. Note that it does not mean that the parameter can be omitted from the lifted action, as it is essential for defining action preconditions. On the labeled transition system, however, all labels of the corresponding grounded actions that differ only in the room parameter can be safely collapsed into one label, achieving a smaller set of labels.

It is no coincidence that discrete action sets of small sizes are also favored by reinforcement learning (RL) approaches. Choosing from a large collection of mostly irrelevant actions in a state can be detrimental to model-free methods (Huang and Ontañón 2022). Most RL benchmarks have only a small number of actions, e.g., Atari benchmarks have at most 18 actions, representing all possible transition labels (Nelson 2021). When planning problems are cast as Markov Decision Processes (MDPs), great care is taken in defining small label sets (Silver and Chitnis 2020; Fern, Yoon, and Givan 2006; Dzeroski, Raedt, and Driessens 2001). In PDDL-Gym (Silver and Chitnis 2020), the label sets are manually crafted by identifying a subset of lifted action parameters that are *inessential* for distinguishing two labels in a state. For example, the `?r:room` parameter from our Gripper example is manually identified as inessential.

In this work, we propose automating this manual process, exploring ways of automatically reducing action labels in classical planning domains. For that, we characterize a *valid* label reduction for classical planning tasks and propose a way to automatically obtain such a reduction. Focusing on the reduction of action parameters, we show how *lifted mutex groups* (Fišer 2020) can be leveraged to automatically identify the inessential parameters of the actions effectively, essentially automating the manual process of Silver and Chitnis (2020). Our contributions, however, have a wider scope. We formally define the problem of obtaining a parameter seed set and propose to solve this problem by translating it to a delete-free planning task, proving that the solution obtained is a valid label reduction. Then we empirically evaluate our approach on 14 IPC domains and 10 hard-to-ground domains (Lauer et al. 2021; Haslum 2011; Matloob and Soutchanski 2016) and show that it achieves a significant reduction in action labels. Finally, we demonstrate the benefits of our approach on two use cases, RL and lifted successor generation (Corrêa et al. 2020). We empirically show that the label reduction can help in both cases: it significantly improves the sample efficiency of standard RL agents and speeds up the time to generate applicable ground action in lifted successor generation.

## 2 Preliminaries

In this section, we first introduce the necessary classical planning notations and then describe lifted mutex groups

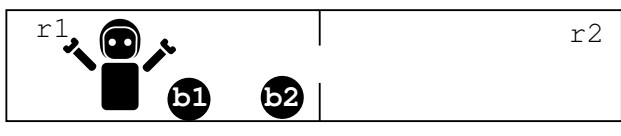

r1               r2

b1   b2

(a)

Gripper task $\Pi = \langle \mathcal{L}, \mathcal{O}, I, G \rangle$.
- Language $\mathcal{L}$ includes:
  - objects $\mathcal{B}$ : r1, r2, b1, b2, g1, g2
  - types $\mathcal{T}$ : room, ball, gripper
  - variables $\mathcal{V}$ : ?r, ?b, ?g, ?f, ?t
  - predicates $\mathcal{P}$ : at_robby, at, free, carry
- Initial state $I$ is: {at(b1,r1), at(b2,r1),
    at_robby(r1), free(g1), free(g2)}
- Goal $G$ is: {at(b2,r2)}
- Actions $\mathcal{O}$ consists of:
  - move  : params {?f : room, ?t : room}
    - : pre {at_robby(?f)}
    - : add {at_robby(?t)}
    - : del {at_robby(?f)}
  - pick  : params {?b : ball, ?r : room, ?g : gripper}
    - : pre {at(?b,?r), at_robby(?r), free(?g)}
    - : add {carry(?b,?g)}
    - : del {at(?b,?r), free(?g)}
  - drop  : params {?b : ball, ?r : room, ?g : gripper}
    - : pre {carry(?b,?g), at_robby(?r)}
    - : add {at(?b,?r), free(?g))}
    - : del {carry(?b,?g)}

(b)

Figure 1: A running example of Gripper task, (a) an initial state with b1, b2, and robot in room r1, and (b) a normalized PDDL task.

that we use in our proposed approach.

## 2.1 PDDL Task

We consider normalized PDDL tasks without axioms and conditional effects (Helmert 2009). A *normalized PDDL task* $\Pi = \langle \mathcal{L}, \mathcal{O}, I, G \rangle$ is defined over a first-order language $\mathcal{L}$, a finite set of *lifted actions* $\mathcal{O}$, an *initial state specification* $I$, and a *goal specification* $G$.

A *first-order language* $\mathcal{L} = \langle \mathcal{B}, \mathcal{T}, \mathcal{V}, \mathcal{P} \rangle$ consists of a finite number of objects ($\mathcal{B}$), types ($\mathcal{T}$), variables ($\mathcal{V}$), and predicates ($\mathcal{P}$). The association between types and objects is defined by a function $\mathcal{D} : \mathcal{T} \mapsto 2^{\mathcal{B}}$. $\mathcal{T}$ contains a special default type $t_0$. Every object is associated with this default type such that $\mathcal{D}(t_0) = \mathcal{B}$. Every pair of types $t_i, t_j \in \mathcal{T}$ satisfy one of the following conditions, either $\mathcal{D}(t_i) \subseteq \mathcal{D}(t_j)$ or $\mathcal{D}(t_i) \supseteq \mathcal{D}(t_j)$ or $\mathcal{D}(t_i) \cap \mathcal{D}(t_j) = \emptyset$. $\mathcal{V}$ is a finite set of variable symbols such that each variable is associated with a type in $\mathcal{T}$. All variables are represented with a prefix "?", for example, ?v. A pair of object and variable (o, ?v) are *compatible* if o $\in \mathcal{D}(t_v)$, where $t_v$ is the type of variable $v$. A predicate in $\mathcal{P}$ has fixed arity and each argument is associated with a type in $\mathcal{T}$. An *atom* is a predicate symbol followed by a parenthesized list of arguments, predicate(term₁, term₂, ⋯) where term$_i$ can be a variable or object. For an atom (or a set of atoms) $\alpha$,

$free(\alpha) \subseteq \mathcal{V}$ denotes a set of variables in the atom (or the set). If $free(\alpha) = \emptyset$ then $\alpha$ is called ground atom; otherwise it is called lifted atom. A lifted atom is grounded by replacing every variable with a compatible object. If lifted atoms $\alpha$ and $\alpha'$ have the same predicate and the types of all the terms in $\alpha$ are subsets of types of respective terms in $\alpha'$, we say that $\alpha$ is a *subset* of $\alpha'$. That is, $p(?a, ?b) \sqsubseteq p(?a', ?b')$, if $\mathcal{D}(t_a) \subseteq \mathcal{D}(t_{a'})$ and $\mathcal{D}(t_b) \subseteq \mathcal{D}(t_{b'})$. A *literal* is an atom or negation of an atom.

The initial state specification $I$ is a conjunction of ground atoms with fluent predicates (that can change over time). The goal specification $G$ is a conjunction of ground atoms or their negations. A *lifted action* $o = \langle head, cost, pre, add, del \rangle$ in $\mathcal{O}$ consists of the atom $head(o)$, indicating the name and the parameters of the action, an optional $cost(o)$, indicating the cost of performing that operation, the preconditions $pre(o)$, the add-effects $add(o)$, and the delete-effects $del(o)$, each is a conjunction of literals over $\mathcal{L}$. For each action $o$, the set of action parameters $params(o)$ is defined as $free(pre(o)) \cup free(add(o)) \cup free(del(o))$. Actions with empty parameter sets are called *ground* actions. Otherwise, an action can be grounded by replacing parameters with compatible objects in the domain.

The set of all ground actions is denoted by $\mathcal{O}_\downarrow$. By $o_\downarrow(P/\theta)$ we denote a set of ground actions induced by assigning objects $\theta$ to parameter subset $P$ and grounding the remaining parameters with all the compatible objects. In the gripper example, the ground action set of the lifted action $o = $ pick(?b, ?r, ?g) induced by the assignment {?b/b1, ?g/g1} is $o_\downarrow(\{?b/b1, ?g/g1\}) = $ {pick(b1,r1,g1), pick(b1,r2,g1)}, where the parameters ?b and ?g are replaced with the objects mentioned in the assignment but the parameter ?r is replaced with all room objects, {r1, r2}.

A *state* $s$ assigns values TRUE and FALSE to all ground atoms with *fluent* predicates. The *initial state* $s_0$ of the task assigns value TRUE to all atoms occurring in $I$, and FALSE to all other fluent ground atoms. A ground action $o$ is *applicable* in state $s$ if $s \models pre(o)$, that is, the preconditions of $o$ are satisfied in the state $s$. A ground atom $\alpha$ is TRUE in the successor state if and only if either it has been TRUE in $s$ and $\alpha \notin del(o)$ or $\alpha \in add(o)$. A *plan* for the task is a sequence of ground actions whose subsequent application leads from $s_0$ to some state $s_*$ with $s_* \models G$.

## 2.2 Lifted Mutex Groups

A *mutex group* is a set of mutually exclusive ground atoms $M$, of which at any given (reachable from $I$) state $s$ at most one can be TRUE. That is, for any reachable state $s$, $|M \cap s| \leq 1$ or equivalently $|\{\alpha \mid s \models \alpha, \alpha \in M\}| \leq 1$. For example, in the gripper domain, {at(b1,r1), at(b1,r2)} is a mutex group as, in any given state, ball b1 can only be in one of the rooms. Any subset of a mutex group is also a mutex group. A *lifted mutex group* (LMG) is a set of lifted atoms that produces a mutex group when grounded. Formally, a lifted mutex group is defined using an *invariant candidate*.

An *invariant candidate* is a tuple $c = \langle v^f, v^c, \mathcal{A} \rangle$ where $v^f(c)$ ($v^c(c)$) is a finite set of fixed (counted) variables (il-

lustrated in the example below) and $\mathcal{A}(c)$ is a finite set of atoms such that all the variables of the atoms are present in either $v^f(c)$ or $v^c(c)$, i.e. $free(\mathcal{A}(c)) = v^f(c) \cup v^c(c)$ and $v^f(c) \cap v^c(c) = \emptyset$. For example, consider an invariant candidate $c = \langle \{?b\}, \{?r\}, \{\texttt{at(?b,?r)}\} \rangle$. Different groundings of fixed variables $v^f(c) = \{?b\}$ generate different sets of ground atoms and different grounding of counted variable $v^c(c) = \{?r\}$ generates ground atoms within each set. We denote the ground atom set with down arrow $_\downarrow$. One of the ground atom sets for $\{?b/b1\}$ is $c_\downarrow(?b/b1) = \{\texttt{at(b1,r1),at(b1,r2)}\}$ and another ground set for $\{?b/b2\}$ is $c_\downarrow(?b/b2) = \{\texttt{at(b2,r1),at(b2,r2)}\}$.

An invariant candidate is called a lifted mutex group if all of its ground atom sets are mutex groups, that is, for any reachable state $s$ and assignment $\{v^f(c)/x\}$, $|\{a \mid s \models a, a \in c_\downarrow(v^f(c)/x)\}| \leq 1$. An LMG with no fixed variable can only generate one ground mutex group. For example, $\langle \emptyset, ?r, \{\texttt{at\_robby(?r)}\} \rangle$ only induces ground atoms set $\{\texttt{at\_robby(r1),at\_robby(r2)}\}$. Fišer (2020) provides a method to identify the set of LMGs for a PDDL task. Since an LMG with multiple atoms can be split into multiple LMGs with a single atom each, for simplicity, in this paper we assume each LMG has only one atom.

## 3 Label Reduction

A planning task can be represented as a labeled transition system, where labels are operations that can be executed in states. These transition labels are identified by the $head(o)$ for ground actions. For example, $\texttt{pick(b1,r1,g1)}$ is a label for the action that picks the ball $\texttt{b1}$ from room $\texttt{r1}$ in the gripper $\texttt{g1}$. *A label set $L$ consists of a unique label for each ground action in $\mathcal{O}_\downarrow$.* The label set size increases exponentially in the number of objects. This work aims to reduce the size of the label set $L$. We do so by identifying an assignment of labels to planning actions such that it generates a smaller label set $L'$ while producing an equivalent transition system. We capture this requirement by specifying the criteria for a *valid label reduction*. A label reduction is *valid* if it assigns distinct labels to any two ground actions that can be applied in the same reachable state. For example, actions $\texttt{pick(b1,r1,g1)}$ and $\texttt{pick(b2,r2,g1)}$ cannot be applied in the same state as the gripper $\texttt{g1}$ cannot be in two different rooms in the same state. Thus, assigning the same label to both would be valid. But $\texttt{pick(b1,r1,g1)}$ and $\texttt{pick(b2,r1,g2)}$ can be applied in the same state, and hence cannot be assigned the same label.

**Definition 1** *A label reduction function $\psi : L \mapsto L'$ is* valid *if any two distinct ground action labels $head(o_1), head(o_2) \in L$ that are applicable in the same reachable state ($s \models pre(o_1) \wedge s \models pre(o_2)$) are assigned distinct labels, that is $\psi(head(o_1)) \neq \psi(head(o_2))$.*

This definition ensures that any two actions that are applicable in the same state are distinguishable. For each reduced label, the set of corresponding actions must include at most one applicable action for each reachable state. Noticing the

resemblance to predicate mutex groups, we call such action sets *applicable action mutex groups*.

**Definition 2** *A set of ground actions $\mathcal{O}'$ is an* applicable action mutex group *(AAMG) if for any reachable state $s$, $|\{o \mid s \models pre(o), o \in \mathcal{O}'\}| \leq 1$.*

Naturally, any subset of an AAMG is also an AAMG, and any set of actions of size 1 is an AAMG. A partitioning of actions into AAMGs defines a valid label reduction, and vice versa, a valid label reduction defines a partitioning of actions into AAMGs. While one can seek to find the smallest possible valid label reduction, that might require generating the set of all ground actions. To avoid the tedious grounding process, we focus on finding AAMGs for lifted actions.

We find AAMGs for each lifted action separately, by reducing its parameters. For example, consider a lifted action $o = \texttt{pick(?b,?r,?g)}$, as a robot can only be in one specific room in any state, only one specific assignment to $?r$ is satisfiable in any state. So one possible set of AAMGs can be obtained by defining partial grounding of action $o$ on the subset of parameters obtained after removing $?r$. That is $o_\downarrow(\{?b/b, ?g/g\}) \mid \quad \forall b, \ g \in \mathcal{B}\} = \{\{\texttt{pick(b1,r1,g1),pick(b1,r2,g1)}\}, \{\texttt{pick(b1,r1,g2),pick(b1,r2,g2)}\}, ...\}$.

A partial grounding of parameter subset ($X \subseteq params(o)$) of a lifted action $o$ induces sets of ground actions where each set corresponds to a particular assignment of objects to parameter subset $X$. Thus, we want to identify a *subset of parameters* ($X$) such that any assignment ($c$) to this subset results in the ground action set ($o_\downarrow(X/c)$) being an AAMG (like the subset $\{?b, ?g\}$ in the above example). Note that LMGs have a similar property. Any assignment to their fixed variables results in a ground atom set being a mutex group. Next, we show how LMGs can be used to identify the required parameter subset.

**Theorem 1** *Given a lifted action $o$ and a lifted mutex group $l = \langle v^f(l), v^c(l), \{\alpha\} \rangle$, if $p \sqsubseteq \alpha$ for some $p \in pre(o)$, then any assignment $c$ to $X = params(o) \setminus v^c(l)$ [1] results in $o_\downarrow(X/c)$ being an AAMG.*

**Proof:** Given an assignment $v^f(l)/c$, any state $s$ can only satisfy at most one of the ground atoms from the mutex group $l_\downarrow(v^f/c)$ (from the definition of LMG). Consequently, as $p \in pre(o)$ and $p \sqsubseteq \alpha$, the state can satisfy at most one of the preconditions of the ground actions in the set $o_\downarrow(X/c)$. Hence, $o_\downarrow(X/c)$ is an AAMG. ∎

We call an LMG $l$ *relevant* to a lifted action if an atom $p$ in the precondition satisfies $p \sqsubseteq \alpha$, where $\alpha \in \mathcal{A}(l)$. The parameters from set $v^c(l)$ of a relevant LMG need not be included in $X$. Given the assignment to $v^f(l) \subseteq params(o)$ the LMG $l$ guarantees a unique assignment to parameters $v^c(l)$. Once the assignment to these parameters ($v^f(l) \cup v^c(l) \subseteq params(o)$) are identified, another LMG $l'$ could now be used to identify the assignment to parameters $v^c(l')$ and hence $v^c(l')$ can also be removed from $X$. Essentially, we can leverage multiple LMGs to further reduce the

---

[1] We assume that the variables of the LMG $l$ match the ones of the precondition atom $p$.

subset $X$. Formally, this corresponds to the following problem, which we call *parameter seed set*:

**Input:** A lifted action $o$ with parameters $params(o)$ and a set of *relevant* lifted mutex groups $L$.
**Find:** A subset $X \subseteq params(o)$ of parameters s.t. $\exists X_1, \ldots X_k$ with (i) $X = X_1 \subseteq X_2 \subseteq \ldots \subseteq X_k = params(o)$, and (ii) $X_{i+1} = X_i \cup v^c(l)$ for some $l \in L$ s.t. $v^f(l) \subseteq X_i$.

Any assignment of objects to the parameter seed set $X$ will result in a unique assignment to all the remaining parameters of $o$ for any reachable state.

**Theorem 2** *Let $o$ be a lifted action over parameters $params(o)$ and $X$ be a solution to the parameter seed set problem above. Any assignment $c$ of objects to $X$ results in $o_\downarrow(X/c)$ being an AAMG.*

**Proof:** Let $X_1 \subseteq X_2 \subseteq \ldots \subseteq X_k = params(o)$ and let $l_1, \ldots, l_{k-1}$ be lifted mutex groups such that $v^f(l_i) \in X_i$ and $X_{i+1} = X_i \cup v^c(l_i)$. Then, each $X_i$ is a solution to the parameter seed set problem. We prove the claim by induction over the number of lifted mutex groups starting from $k$. The base claim of $o_\downarrow(X_k/x)$ (one lifted mutex group) is AAMG results from Theorem 1. We assume that $o_\downarrow(X_{i+1}/\hat{c})$ is an AAMG for any assignment $\hat{c}$ to $X_{i+1}$ and prove that $o_\downarrow(X_i/\tilde{c})$ is an AAMG for any assignment $\tilde{c}$ to $X_i$. Since $l_i = \langle v^f(l_i), v^c(l_i), \mathcal{A}(l_i) \rangle$ is a lifted mutex group with $v^f(l_i) \subseteq X_i$, we have that $l_{i\downarrow}(X_i/\tilde{c})$ is a mutex group. Let $o_1, o_2$ be two ground actions in $o_\downarrow(X_i/\tilde{c})$. If both $o_1$ and $o_2$ belong to $o_\downarrow(X_{i+1}/\hat{c})$, we are done. Otherwise, assume $o_1$ in $o_\downarrow(X_{i+1}/c_1)$ and $o_2$ in $o_\downarrow(X_{i+1}/c_2)$, where $c_1$ and $c_2$ agree on $X_i$ but differ on $X_{i+1} \setminus X_i$. However, since $X_{i+1} = X_i \cup v^c(l_i)$, we have $X_{i+1} \setminus X_i \subseteq v^c(l_i)$, making $c_1$ and $c_2$ mutually exclusive. Thus, $o_\downarrow(X_i/\tilde{c})$ is an AAMG. ∎

Different parameter seed sets $X$ correspond to different AAMGs. To find the smallest possible label set $L'$, we want to minimize the number of AAMGs and therefore we are looking for a seed set $X$ with a minimum possible total number of assignments. This can be expressed as

$$\operatorname*{argmin}_X \prod_{x \in X} |\mathcal{D}(x)|.$$

As the objective is not linear, we can use an equivalent one instead: $\operatorname*{argmin}_X \sum_{x \in X} \log(|\mathcal{D}(x)|)$.

The parameter seed set problem is NP-Complete. For the lack of space, the proof, by reducing the bounded parameter seed set decision problem to a seed set decision problem (Gefen and Brafman 2011), is deferred to the supplementary material. To solve the parameter seed set problem, we cast it as a (delete-free) STRIPS planning task with operation costs. We first find a set $L$ of relevant LMGs. Then, for each lifted action $o$ we define a separate planning task $\Pi_o = \langle \mathcal{L}_o, \mathcal{O}_o, I_o, G_o \rangle$, where

- Language $\mathcal{L}_o$ contains a single predicate mark and an object for each parameter in $params(o)$.
- The set $\mathcal{O}_o$ consists of two types of actions

1. $\text{seed}_x$ actions are defined for each parameter $x \in params(o)$ as $\text{seed}_x := \langle \text{seed}_x, log(|\mathcal{D}(x)|), \emptyset, \{\text{mark}(x)\}, \emptyset \rangle$
2. $\text{get}_l$ actions are defined for each relevant LMG $l$ as $\text{get}_l := \langle get_l, 0, \{\text{mark}(x) \,|\, x \in v^f(l)\}, \{\text{mark}(y) \,|\, y \in v^c(l)\}, \emptyset \rangle$.

- Initial state $I_o = \emptyset$
- Goal state $G_o = \{\text{mark}(x) \,|\, \forall x \in params(o)\}$.

The action $\text{seed}_x$ marks parameter $x \in params(o)$ as an element of the seed set. Action $\text{get}_l$ indicates that a unique assignment for the parameters $x \in v^c(l)$ can be identified if all parameters $y \in v^f(l)$ are known. Therefore, the parameters $v^c(l)$ can be reduced. A plan for $\Pi_o$ corresponds to a sequence of seed and $\text{get}_l$ actions. The parameters marked by seed actions form the seed set, while others are reduced.

**Theorem 3** *For a plan $\pi$ of $\Pi_o$, $X_\pi = \{c \,|\, \text{seed}_c \in \pi\}$, is a solution to the parameter seed set problem of $o$.*

**Proof:** Let $\pi$ be a plan for $\Pi_o$ (assume there are no redundant repetitions of actions in $\pi$). Since seed actions have no preconditions, assume these actions come before $\text{get}_l$ actions, and let $\pi = \pi_s \pi_g$ denote the partition of $\pi$ into the two sequences of seed and $\text{get}_l$ actions, respectively. Let $s_1$ be the state resulting from applying $\pi_s$ in the initial state $I_o$ and $s_1, \ldots, s_k$ be the sequence of states along $\pi_g$ applied to $s_1$. Then, we have (i) $s_1 \subseteq s_2 \subseteq \ldots \subseteq s_k$ and $s_k = \{\text{mark}(x) \,|\, x \in params(o)\}$, as well as (ii) $s_{i+1} = s_i \cup add(\text{get}_l) = s_i \cup \{\text{mark}(y) \,|\, y \in v^c(l)\}$ for some $\text{get}_l$ with $pre(\text{get}_l) = \{\text{mark}(x) \,|\, x \in v^f(l)\} \subseteq s_i$. Denoting the parameters of $o$ marked in the state $s$ by $\Gamma(s) = \{x \,|\, \text{mark}(x) \in s\}$, we get that $X_\pi = \Gamma(s_1)$. ∎

The cost of a plan $\pi$ is $\sum_{\text{seed}_x \in \pi} \log(|\mathcal{D}(x)|)$, and therefore a cost-optimal plan will correspond to a parameter seed set with a minimal possible total number of assignments. To summarize, we find a parameter seed-set $X$ for each lifted action such that assigning objects to $X$ will result in a set of ground actions that is an AAMG. Hence, all the ground actions in that set can be assigned the same label. This reduces the size of the label set $L$.

## 4 Experiments

Our experimental evaluation is split into three parts. First, we check whether our approach is able to reduce the size of the transition label set and whether the reduction is substantial. The next two parts evaluate the utility of our approach. We test whether our reduction can translate into improved performance in two use cases: learning reinforcement learning policies and lifted successor generation.

### 4.1 Reduction in the Label Sets

We compare the size of label sets, obtained with and without the proposed reduction, on a representative set of 14 STRIPS domains from various IPC (using the typed versions where available) and 10 hard-to-ground (HTG) domains. We use the Fast Downward (Helmert 2006) planning system translator to ground the lifted actions. To infer the lifted mutex groups, we use the implementation by Fišer (2020) and

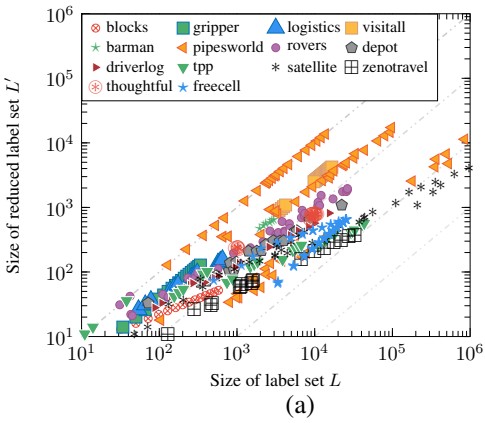 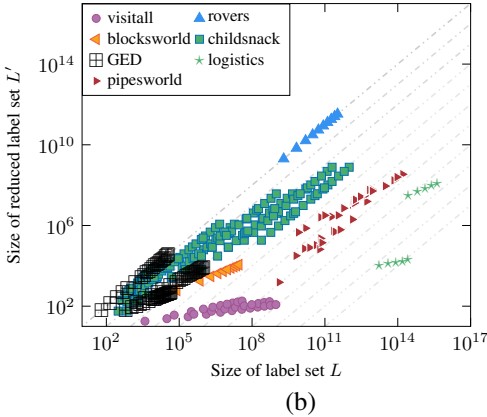

(a)                                                (b)

Figure 2: Comparison of label set sizes on (a) $14$ IPC STRIPS domains and (b) $7$ HTG domains.

to solve the *parameter seed set* planning task we use the Fast Downward planner with A* search. The parameter seed set planning problem described in the previous section uses real-valued costs for the seed actions. However, the Fast Downward planner only allows integer costs. So we scale the real-valued costs of the seed action in our experiments.

Figure 2 compares the size of the label sets $L'$ and $L$, obtained with and without the reduction resp. Figure 2a presents the reduction on each PDDL problem instance for IPC domains and Figure 2b on 7 out of 10 HTG domains (the remaining 3 are available in the supplementary material). Both axes are log-scale. Points below the diagonal indicate instances where the reduced label set is smaller than the original one. The distance from the diagonal indicates the significance of the reduction. Gray dashed lines below the diagonal represent the order of magnitude of the reduction. *Our experimental results show a substantial reduction of the label set in most problem instances, going up to 2 orders of magnitude on IPC problems and up to 10 orders of magnitude on hard-to-ground domains.*

Table 1 summarizes the number of lifted actions that were reduced by our approach. It also presents the mean and max number of non-seed parameters found in the lifted actions, i.e., $|params(o)| - |X|$. Each row of the table represents a domain, aggregating results over the instances of that domain. We were able to find non-seed parameters in all 14 IPC domains and all but 2 HTG domains. More than 3 parameters were reduced for some lifted actions in *thoughtful, pipesworld, tpp, freecell, zenotravel, visit all, childsnack,* and *OS-alkene* domains. 2 IPC domains and 3 HTG domains have actions with $100\%$ reduction, that is all the parameters of some actions were deemed inessential. Note that the number of reduced parameters (in Table 1) is not necessarily proportional to the reduction in the label set (in Figure 2). Nevertheless, the number of reduced parameters indicates the importance of parameter reduction. The computation time was between $0.25$ and $1.73$ seconds for the IPC domains and between $0.26$ and $4.69$ seconds for the HTG domains.

## 4.2 Learning RL Policies

An MDP $\mathcal{M} = \langle S, A, P, R \rangle$ contains a set of states $S$, a set of actions $A$, a transition probability distribution $P : S \times S \times A \mapsto [0, 1]$, and a reward function $R : S \mapsto \mathbb{R}$. When a PDDL task $\Pi$ is cast as an MDP $\mathcal{M}$, the set of states $S$ are defined as the set of all states reachable from $I$ of $\Pi$, the action set $A$ is defined as the set of labels that is composed of a unique label for each of the ground actions, the probability distribution $P$ is defined to respect the state-transition in the PDDL actions, and the reward function $R$ is defined as some positive integer when $s \models G$ and 0 otherwise. In practice, for each of the ground actions, the head of the ground action $head(o)$ is assigned as the unique label.

To evaluate the advantage of reducing the label set size in planning as RL, we cast the PDDL task as an MDP with two different action spaces: 1) **All**: default action space with all ground actions, with $head(o)$ as unique labels, 2) **Reduced**: reduced action space with one label for each AAMG and compare the learning curves of RL policies. We focus on 4 classical planning domains, *ferry, gripper, blocks,* and *logistics*. Since our aim is to evaluate the advantage of reducing the action space, and not the generalization of policies, we fix the number of objects in each domain. We generate 500 unique pairs of initial and goal states in each domain. Of these, 250 pairs were used in training and the remaining were set aside for evaluation. Inspired by the work of Gehring et al. (2022), we use domain-independent planning heuristic, $h^{\text{FF}}$, as a dense reward function and use their code to convert the PDDL problem to an RL environment. We employ the Double DQN implementation from the ACME RL library (Hoffman et al. 2020) to learn a state-action value function and apply a greedy policy $\pi(s) = \max_a Q(s, a)$ in our evaluation.

Figure 3 shows learning curves aggregated over 5 runs with different random seeds. For *ferry* and *gripper* domains (Figure 3 a and b), the reduction of action labels improves sample efficiency by as many as $300,000$ steps. In *blocks* and *logistics* domains (Figure 3 c and d), the baseline without the label reduction was not able to learn a policy. With a reduced label set, the training becomes feasible.

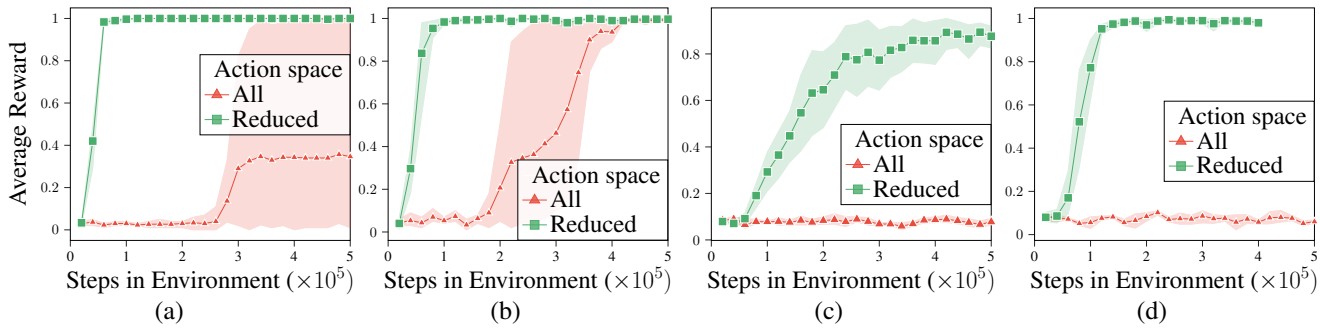

Figure 3: Learning curve in the (a) ferry, (b) gripper, (c) blocks, and (d) logistics; with and without action label reduction.

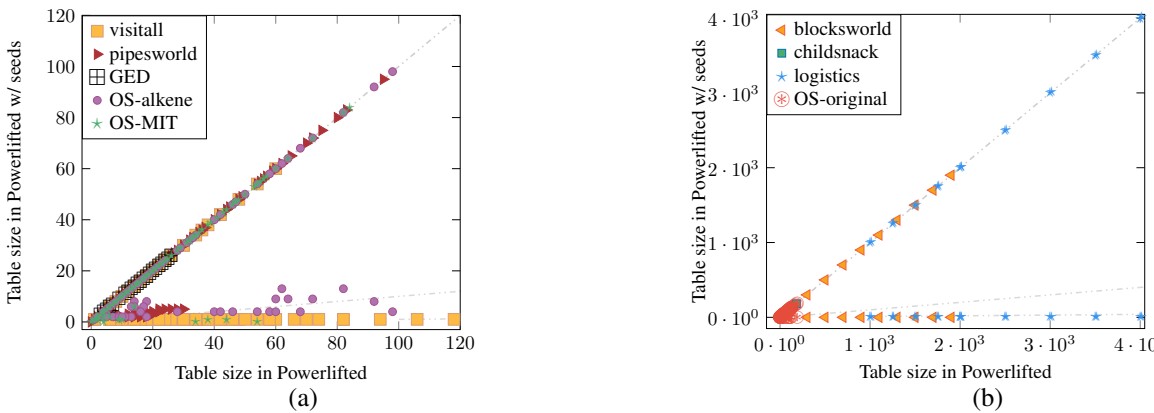

Figure 4: Comparison of table sizes before the query is performed. We split HTG domains into two plots for readability.

It is clear from these plots that reducing the action label sets yields significant gains in terms of sample efficiency. One possible reason that explains these results is the reduction of invalid actions achieved in these environments. As our approach reduces the action labels while maintaining all the valid actions, the number of invalid actions is reduced. This phenomenon of deep RL agents showing performance improvement upon reduction of invalid actions is studied in Huang and Ontañón (2022). Well aware of this phenomenon, researchers invest significant effort to manually identify small action spaces in planning domains (Silver and Chitnis 2020; Fern, Yoon, and Givan 2006; Dzeroski, Raedt, and Driessens 2001) and code state-dependent action functions in RL (Boutilier et al. 2018; Huang and Ontañón 2022; Bamford and Ovalle 2021). We automated this manual process for planning domains. Our results show that even in small-scale problems (with 4–7 objects) label reduction is beneficial. In large problems (with many objects) our approach can provide tremendous leverage for training RL algorithms, reducing label set by orders of magnitude.

## 4.3 Lifted Successor Generation

While planning tasks are often represented in PDDL, using first-order representations, most planners use a propositional or a multi-valued grounded representation (Bäckström and Nebel 1995). The lifted successor generator by Corrêa et al. (2020) works directly on a lifted level to generate

successor states using database techniques (Ullman 1989). A state is represented as a database. Each predicate has a table with the number of columns according to the predicate arity. Each fact in the state forms a row in the table. With this state representation, the task of identifying the applicable actions is equivalent to a join query evaluation. Consider a planning task $\Pi = \langle \mathcal{L}, \mathcal{O}, I, G \rangle$ over a first-order language $\mathcal{L} = \langle \mathcal{B}, \mathcal{T}, \mathcal{V}, \mathcal{P} \rangle$. A state $s$ is a database $D(s) = \langle \mathcal{B}, \{R_{P,s} | P \in \mathcal{P}\} \rangle$ with objects $\mathcal{B}$ as domain and finite set of relations over these objects. The relation $R_{P,s}$ contains all ground atoms of predicate $P$ in state $s$ as tuples. The set of ground actions applicable in $s$ for a lifted action $o \in \mathcal{O}$ is identified by the conjunctive query

$$Q(params(o)) :\text{--} R_{P_1,s}, \cdots, R_{P_n,s} \text{ where } P_i \in pre(o).$$

The query result provides tuples of object assignments to the action parameters, which define the ground actions that are applicable in the state.

There are several possible ways of exploiting the additional information of the seed parameters for speeding up join computation. The complexity of the query evaluation is measured in terms of the input and output size of the query. With our seed parameter set, the input and the output of the query can be modified and improvement can be achieved in computation time. To modify the output, one can query only for the seed parameters and derive the assignments for non-seed parameters using the sequence of lifted mutex groups.

| Domain | # reduced actions | non-seed parameters | |
| --- | --- | --- | --- |
| | | max % (#) | mean % (#) |
| IPC domains | | | |
| blocks | 3/4 | 100.0% (1.00) | 50.00% (0.75) |
| barman | 11/12 | 66.67% (3.00) | 41.94% (1.92) |
| driverlog | 6/6 | 66.67% (2.00) | 47.22% (1.50) |
| thoughtful | 20/21 | 100.0% (6.00) | 73.03% (3.24) |
| gripper | 3/3 | 66.67% (2.00) | 50.00% (1.33) |
| pipesworld | 6/6 | 74.57% (6.12) | 65.69% (5.26) |
| pipesworld (no t.) | 6/6 | 71.43% (5.00) | 59.81% (3.87) |
| pipesworld (no s.) | 4/4 | 68.89% (7.60) | 65.83% (6.88) |
| tpp | 4/4 | 69.52% (4.87) | 60.48% (3.90) |
| freecell | 10/10 | 80.00% (5.00) | 65.29% (3.30) |
| logistics | 6/6 | 66.67% (2.00) | 55.95% (1.76) |
| rovers | 8.62/9 | 77.08% (2.88) | 46.50% (1.73) |
| satellite | 5/5 | 68.52% (2.08) | 51.99% (1.46) |
| visitall | 1/1 | 50.00% (1.00) | 50.00% (1.00) |
| depot | 5/5 | 50.00% (2.00) | 46.67% (1.80) |
| zenotravel | 5/5 | 77.50% (4.10) | 62.23% (2.68) |
| HTG Domains | | | |
| visitall-3dim | 3/3 | 75.00% (3.00) | 75.00% (3.00) |
| visitall-4dim | 4/4 | 80.00% (4.00) | 80.00% (4.00) |
| visitall-5dim | 5/5 | 83.33% (5.00) | 83.33% (5.00) |
| blocksworld | 3/4 | 100.0% (1.00) | 50.00% (0.75) |
| GED | 11/14 | 100.0% (3.00) | 61.90% (1.50) |
| GED-split | 19/21 | 100.0% (2.00) | 73.81% (1.38) |
| GED-positional | 0/3 | 0.00% (0.00) | 0.00% (0.00) |
| pipesworld (no s.) | 4/4 | 68.00% (7.26) | 64.10% (6.68) |
| rovers | 0/9 | 0.00% (0.00) | 0.00% (0.00) |
| childsnack_parsize1 | 2.5/4 | 63.33% (3.17) | 27.29% (1.17) |
| childsnack_parsize2 | 3/4 | 77.78% (4.67) | 36.11% (1.92) |
| childsnack_parsize3 | 3/4 | 80.95% (5.67) | 37.95% (2.42) |
| childsnack_parsize4 | 3/4 | 83.33% (6.67) | 39.17% (2.92) |
| logistics | 6/6 | 83.33% (2.50) | 65.97% (2.08) |
| OS-MIT | 15.22/52 | 44.41% (4.39) | 7.91% (0.81) |
| OS-alkene | 12/12 | 67.36% (7.11) | 37.28% (3.81) |
| OS-original | 14.85/52 | 45.74% (4.65) | 7.14% (0.76) |

Table 1: Summary of actions reduced by our approach. Column 2 shows the number of reduced/total lifted actions. Columns 3 & 4 present the maximum & mean of the percent (number) of reducible parameters per action, aggregated over problems in that domain.

In our preliminary experiment, we modify the procedure of Powerlifted planner (Corrêa et al. 2020) by preprocessing the tables, hence modifying the input size. Before querying, we join the precondition tables with the corresponding lifted mutex group table, over non-seed parameters. This allows us to reduce the size of the tables in the query. Figure 4 shows the difference in the size of the tables before the query is performed. The X-axis presents the size of the table in Powerlifted and the Y-axis presents the size of the table in Powerlifted with seeds. So, a point below the diagonal indicates reduced sizes. There are 4914, out of 36097 tables, that undergo size reduction, and often a significant one. In 478 out of 811 problems where the join is performed, at least one table is reduced in size. As known from database literature, reducing the size of the tables can help improve join performance (Ullman 1989). Our initial results (in supplementary material), comparing the time taken to perform the join and to generate applicable actions show that our approach has the potential to save computational cost. Note that we do not modify the existing query evaluation process. Optimizing the join query using the cardinality of the relations can potentially further improve the processing time. However, query optimization is out of the scope of the current work. Further research is needed into additional variants of improving the lifted successor generation to make it beneficial in other domains.

## 5 Related Work

Various approaches have been studied in RL to reduce the action space. Stochastic action sets (Boutilier et al. 2018) and invalid action masking (Huang and Ontañón 2022; Bamford and Ovalle 2021; Kanervisto, Scheller, and Hautamäki 2020) restricts the action selected by an agent to a small subset of actions that are feasible in the given state. This is done by assigning zero probability (or $-\infty$ score) to invalid actions. While the stochastic action sets and invalid action masking define a state-dependent subset of feasible actions, our action reduction is independent of the current state.

Another approach to manage a large number of actions in an MDP is by using factored action spaces (Pazis and Lagoudakis 2011; Geißer, Speck, and Keller 2020; Guestrin, Lagoudakis, and Parr 2002). With factored action space, an action is decomposed into multiple components and represented as either a decision tree or a vector. It is straightforward to convert predicate action space (for example, gripper actions {drop(b1, r2, g1), pick(b2, r1, g2),...}) to a factored action space $(a_0, a_1, \ldots, a_n)$ with $a_0$ denoting the action identifier (for example, drop or pick) and $a_1, \ldots, a_n$, denoting the parameters. Our approach of identifying the parameter seed set can be used to reduce the number of factors in the factored action spaces.

In planning literature, label reduction techniques are used to reduce the number of transition labels in an abstract transition graph (Helmert et al. 2014; Sievers, Wehrle, and Helmert 2014), with the aim to simplify the transition system by creating an equivalence between labeled transitions. Here, the purpose is different: labels of actions that are never applicable together are reduced to the same label while allowing to differentiate between applicable actions in a given state.

## 6 Discussion and Future Work

In this work, we have introduced definitions of a valid label reduction and applicable action mutex groups and have shown the connection between the two. We have presented a method for automatically deriving action label reductions for planning tasks based on action parameter reduction. For that, a parameter seed set problem was introduced, and a solution to the problem was suggested by translating it to delete-free planning. Our experimental evaluation shows a significant reduction in action labels when using our approach, across all tested planning domains. This reduction translates both into improved sample efficiency of standard RL agents and into reduced computation time of identifying

applicable ground actions in lifted planning, the two example use cases.

Our method, however, does not guarantee the optimality of the valid reduction size, even for the restricted case considered in this work. Finding provably minimal size reductions is an interesting topic for future research. Further, we barely touched on the possible benefits of action parameter reduction for classical planning. We have not explored other methods of speeding up successor computation. Finally, exploring the possible benefits of the action parameter reduction for lifted heuristic computation (Corrêa et al. 2021; Lauer et al. 2021) is of great promise for lifted planning.

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
