# OpenReview forum: "On Reducing Action Labels in Planning Domains"
_icaps-conference.org/ICAPS/2023/Workshop/HSDIP — ICAPS HSDIP 2023_

### Official Review · Reviewer_krqn · 2023-04-17
**Good paper on label reduction that spans multiple planning areas from MDPs to lifted planning**

**Rating:** 7
**Confidence:** 4

**Review:**

Incidentally, I already reviewed this paper when submitted to IJCAI and I proposed to accept the paper. My assessment did not change. So, below is pasted my previous review:


The paper introduces a novel method for reducing the number of labels in
transition systems of planning tasks. In contrast to other works focusing on
collapsing labels that are always parallel, this work focuses on the label
reduction over actions that can never be applied in the same state and
therefore using the same name for all of them preserves the structure of
transition system. The proposed method works on the lifted (PDDL) level and it
is experimentally demonstrated that the reduction can lead to a significant
improvement in learning RL policies (where the input PDDL task is transformed
to an MDP). Moreover, it is also shown that this technique could be useful in
lifted classical planning as it improve join-queries corresponding to finding
applicable actions for the given state.


The paper is mostly well written, but the formal parts could be improved (see below).
All proofs seem to be correct and the experimental evaluation is convincing.
The main reason why I think the paper should be accepted is the novelty of the
work that stretches over different planning areas. Frankly, I'm surprised that
such a (conceptually) simple technique can have so profound effect. The
application to RL is straightforward as its performance clearly depends on the
number of actions and it seems the fact that the proposed method works on the
lifted level makes it possible to reduce the number of actions without much
overhead. I also very much like the preliminary work in the context of lifted
planning. The application of the proposed technique there is not obvious, in
my opinion, and it can provide a solid basis for future research of lifted
successor generators, grounding techniques, partial grounding, or other areas
dealing with "hard-to-ground" planning tasks.


Issues:

The paper is mostly well written but the formal
parts, in my opinion, can be improved. Below is an assorted list of my notes:

Why do you use both terms operator and action? It just makes the paper harder
to read. I would suggest to choose one and stick to it throughout the paper
(both are used so it doesn't matter which one is chosen).

What is "fluent predicate" in contrast to "predicate"? The term "fluent" is
also used in connection with ground atoms, but I'm not sure what is meant by
it. It seems to me this term does not carry any useful meaning in the context
of this paper, so I think it can be completely removed.

Using "?" to prefix variables in mathematical notation feels weird to me.
Although I understand it comes from the PDDL syntax, I would suggest to use
standard letters without any prefix, maybe with a different typesetting (or
reserving some letters for denoting variables and other for objects) if it
makes the notation easier to follow.

free(x) needs to be extended also to a set of atoms x. It is defined only for
atoms, but also used for a set of atoms.

Universal quantifiers are missing domains for which they are restricted.

---

### Official Review · Reviewer_h1Y4 · 2023-04-24
**Nice and interesting examination of automated label reduction to make domain-independent planning algorithms more efficient.**

**Rating:** 8
**Confidence:** 3

**Review:**

### Summary
The paper suggests to apply automated label reduction as a preprocess to
classical planning. This is an advancement compared to related work
where this was done by hand on e per-domain basis. Reducing the labels
is relevant to make certain planning approaches more efficient. The
approach proposed in this paper considers mutexes that can be derived
from the PDDL description of a planning task. These mutexes are used to
identify groups of operators such that at most one operator from each
group is applicable in any given state. It is then possible to represent
this group by a single label. In an empirical study it is shown that
reinforcement learning and lifted planning indeed benefit from this kind
of reasoning.

### Scholarship
The write-up is mostly clear and easy to follow. The topic fits well in
the scope of HSDIP as it generalizes domain-dependent practices by a
domain-independent automation. The technique seems novel, at least I am
not aware of any publications of similar approaches. However, I was
personally distracted by the fact that the notion of label reduction
exists in planning (e.g., it is a relevant topic for merge-and-shrink
abstractions) but this is not mentioned before the second to last
section of the paper. I would recommend to move Section 5 "Related Work"
to the front so that readers familiar with the literature get to learn
about the relationships early on.

I was also struggling with the fact that several results are not shown
in the paper but promised to be found in the appendix, but no appendix
was submitted. Furthermore, there would have been enough space to put
some of these results in the main paper, as there is more than one page
left before the page limit.

### Questions
(1) I did not quite understand your discussion of the difference to
label reduction as applied in merge-and-shrink (M&S) abstractions. While
labels in M&S are grouped into equivalence classes, you claim your
approach to be different because you group operators such that never
more than one is applicable in the same state. But isn't this basically
an equivalence relation? For example, \texttt{pick(b1, r1, g1)} and
\texttt{pick(b1, r2, g1)} are grouped together in your approach, and I
would argue they are equivalent as they both have essentially the same
effect (ball in g1 and no longer in r1 or r2, respectively) but are only
applicable in different contexts (states where robby and b1 are in r1 or
r2, respectively).

(2) What would you expect from applying your techniques to grounded
classical planning, e.g., as a preprocessing of Fast Downward? Did you
consider this at any point in your experimental evaluation? Is there a
good reason to not do that?

(3) You mention in several places that the states you consider must be
reachable from the initial state. Why is this relevant? Are there any
problems if this is not the case? And how can you test that efficiently
in practice?

### Conclusion
I like the paper and its premise and therefore suggest to accept it. It
is an interesting read and has a good mix of theory and experimental
results. I ask the authors to make sure the appendices are available
upon acceptance.

### Minor Comments
#### Introduction
- The first sentence in the introduction sounds like PDDL is a
  requirement for planning tasks to induce transition graphs, but this
  is clearly not true.
- Introduction, 1st paragraph: "Figure 1 depicts the ..." --> "Figure 1
  depicts a ..."
- On the one hand, I like the illustration in Figure 1a very much. On
  the other hand, I feel like Figure 1b takes up too much space
  regarding its benefit to the paper (only directly referenced in the
  intro). Moreover, to me it raised the question where the association
  of variables with types is specified, because this does not appear in
  the PDDL description.

#### Preliminaries
- The term "literals" (third paragraph of "PDDL task") is never
  introduced.
- First paragraph of "PDDL task": "$\textit{free}(\alpha) \in
  \mathcal{V}$" --> "$\textit{free}(\alpha) \subseteq \mathcal{V}$"
- Last paragraph of "PDDL task": Your use of $\models$ is not properly
  introduced in my opinion (both for preconditions and goals).
- Use only one definition for mutex groups. Moreover, it is not clear to
  me what $M \cap s$ is supposed to mean, so I would go for the other
  definition.
- I assume a macro is corrupt, namely the third item in the invariant
  candidate definition. (Missing in several other places as well.)
- "An LMG with no ..." --> "A LMG with no ..."
- PDDL task as MDP: There is an unnecessary $\in$ in this paragraph.
- The last part of "Lifted Successor Generation" is not very clear to
  me.

#### Label Reduction
- Proof of Theorem 2: "let $l_1 \dots l_{k-1}$" --> "let $l_1, \dots,
  l_{k-1}$"
- Enumeration list: You define operators as 4-tuples but use 5-tuples
  here. (Costs do not appear in the definition.)

#### Experiments
- Reference missing for Fast Downward (only given for translator). Also,
  Fast Downward offers many planning approaches, which one do you
  consider/use to solve your problems? You should specify that.
- Table 1: Don't use centered alignment, it makes numbers hard to
  compare between individual lines. Use multiple columns if necessary.
- Figures 2 and 4: I recommend to use relative plots where you use % as
  the metric on the y-axis instead of the same as for the x-axis. Doing
  so uses the space better because currently the upper left triangle is
  completely empty of data points. The instances where both numbers have
  the same value would then end up on a horizontal line at the top of
  the plot, while a data point where the label set is reduced by factor
  2 appears right in the middle between the top and bottom border. (Move
  the legend outside the plot if necessary.)
- Second paragraph of "Learning RL policies": "... over 5 runs with
  random seeds." --> "... over 5 runs with different random seeds."
- The planner by Corrêa et al. is called Powerlifted with capital P.

#### Related Work
- "-inf" --> "$-\infty$"

#### Inconsistencies over multiple sections
- For function definitions, you sometimes use $\rightarrow$ and sometimes
  $\mapsto$.
- Most keywords are emphasized using \emph, but sometimes bold font is
  used instead.
- Domain names are sometimes written in italic font, sometimes
  capitalized, sometimes not.

---

### Decision · Program_Chairs · 2023-05-05

**Decision:**

Accept

**Comment:**

Congratulations! We decided to accept this paper to be presented at this year's HSDIP workshop. The reviewers agree that it fits the scope of the workshop and is a relevant and interesting contribution.

Make sure to address the comments made by the reviewers before submitting the camera ready copy. Please also consider adding clarifications regarding the questions raised in the reviews.